# AUTOENCODER WITH DISTRIBUTION PRESERVATION

## ABSTRACT

This paper proposes an improved autoencoder method. On the basis of maintaining the reconstruction accuracy, we introduce a data distribution preservation mechanism to improve the performance of the dimensionality reduction of the model. Traditional autoencoders only focus on the point-to-point distance between the input sample and its reconstruction result, ignoring the preservation of the overall distribution structure of the data. To solve this problem, we introduced the Kernel Mean Embedding (KME) term based on a kernel function with good topological properties into the loss function to measure the difference between the original data distribution and the reconstructed data distribution. This method effectively maintains the topological features and distribution characteristics of the global data. Experimental results show that compared with traditional autoencoders and existing topological autoencoders, our method performs better on multiple datasets, especially in terms of dimensionality reduction quality and structural preservation of latent representations.

## 1 INTRODUCTION

Machine learning methods are playing an increasingly pivotal role in data analysis and knowledge discovery across diverse domains Jordan & Mitchell (2015). However, many traditional algorithms struggle with the so-called "curse of dimensionality" Köppen (2000); Verleysen & François (2005), where performance degrades as the number of features grows, due to sparsity and increased computational complexity Hammer (1962). High-dimensional datasets have become ubiquitous in modern applications, ranging from image representations Krizhevsky et al. (2012), genomics Leek & Storey (2007); Zhang et al. (2025), sensor networks Yick et al. (2008), to text and semantic embeddings Mikolov et al. (2013). While such data often encode rich and informative structures, their high dimensionality poses significant challenges for visualization, clustering, classification, and other downstream tasks.

Dimensionality reduction (DR) addresses this challenge by learning compact, low-dimensional representations that preserve essential structural characteristics of the original data. Classical linear techniques such as Principal Component Analysis (PCA) Abdi & Williams (2010) are widely used but limited in capturing nonlinear manifolds. To overcome these limitations, nonlinear DR methods like t-Distributed Stochastic Neighbor Embedding (t-SNE) Maaten & Hinton (2008) and Uniform Manifold Approximation and Projection (UMAP) McInnes et al. (2018) have gained popularity for their ability to reveal clusters and local structures in visualizations. These methods are often employed as preprocessing steps to facilitate downstream analysis.

Autoencoders (AEs) have emerged as a powerful and flexible framework for nonlinear dimensionality reduction Hinton & Salakhutdinov (2006). By learning an encoder-decoder architecture that maps input data to a low-dimensional latent space and reconstructs it back to the original space, AEs capture salient data features through minimization of reconstruction error. This enables the discovery of meaningful latent representations in an unsupervised manner. However, standard autoencoders primarily focus on pointwise reconstruction fidelity, which may lead to suboptimal latent embeddings where global topological and distributional structures—such as cluster separation, class coherence, or data manifold geometry—are not adequately preserved Dai & Wipf (2019). As a result, the latent space can suffer from overlapping clusters, distorted inter-class distances, and poor interpretability, limiting their effectiveness in tasks such as clustering or visualization.

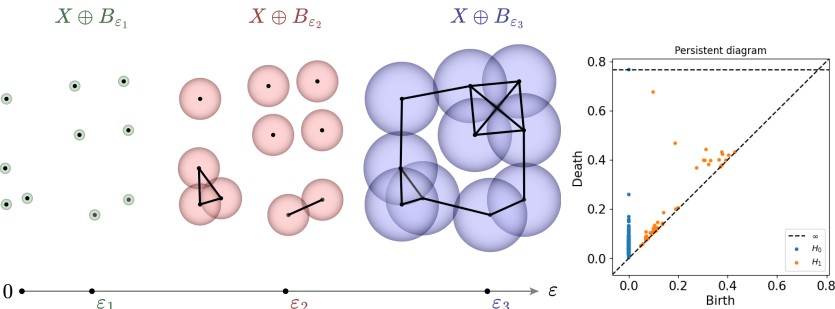

Figure 1: An illustration of persistent homology and persistent diagrams. Given a point cloud, as gradually enlarging a radius $\epsilon$, an increasing sequence of complexes (Vietoris–Rips, etc.) grows; as $\epsilon$ increases, connected components, rings, etc., are "born" and later "die". Persistence diagram is a set of points whose x-coordinate is birth $\epsilon$-values and y-coordinate is death$\epsilon$-values.

To mitigate these issues, recent research has explored integrating additional inductive biases into the autoencoder framework Moor et al. (2020). Topology-preserving autoencoders aim to maintain the intrinsic geometric or topological structure of the data manifold in the latent space. Topological Autoencoders Using persistent homology (PH, An example is shown in Figure 1), a technique from topological data analysis, to calculate topological signatures of both the input and latent space to derive a topological loss term. This has been shown to preserve the topological structure of data before and after dimensionality reduction. However, topological autoencoders aim to preserve topological properties by minimizing the distance between the Persistent Diagrams (PDs) generated before and after dimensionality reduction. While they have implemented some acceleration schemes for this purpose, this still requires the computation of the Vietoris–Rips Complex, a high complexity that limits their application to large-scale data. *Is it possible to quickly compute topological similarity without computing the PH while preserving topological structures?*

Note that the autoencoder only focuses on preserving the relationship between point pairs (input space and reconstruction space), and often neglects global information. This is why using topology can improve its dimensionality reduction effect. Therefore, by jointly preserving the relationship between point pairs and aligning the relationship between distributions, global structural information can be effectively retained. With this as motivation, this paper promotes the preservation of global information by adding distribution loss rather than topological loss. The challenge lies in how to measure the similarity of distributions in different spaces. We employ the kernel mean embedding based Muandet et al. (2017) on a kernel that has been proven to have good topological properties, $\Lambda$-kernelZhang et al. (2023). We prove that this kernel is dimensionality-unbiased [1], and our experimental results show that distribution loss is effective in maintaining global information and topological information without calculating the complex (such as Vietoris–Rips Complex).

Our contributions are threefold:

1. Proposing the notion of similarity between distributions in different spaces and defining the dimensionality-unbiased metric.

2. Proving that the $\Lambda$-kernel is dimensionality-unbiased. while the Gaussian kernel is dimensionality-biased.

3. Proposing a distribution autoencoder (DAE) based on the distribution loss and experimentally demonstrating the superiority of DAE.

## 2 RELATED WORK

Autoencoders are unsupervised learning models Baldi (2012). Based on the backpropagation algorithm Cilimkovic (2015) and optimization methods (such as gradient descent) Ruder (2016), they use the reconstruction error of the input data as supervision to guide the neural network in trying to

---

[1]See Definition in Section 3.

learn a mapping relationship, thereby obtaining a reconstructed output. The embedding layers are often used as a result of dimensionality reduction Wang et al. (2016). Since it does not require labels and is easy to train, autoencoders have been widely used in many fields Chen et al. (2022); Gala et al. (2019); Grassi et al. (2022); Ilnicka & Schneider (2023). However, since autoencoders only consider the loss between points in the input space and the reconstruction space, their effects are often poor because they lose global information. Topological autoencoders (TAEs) combine topological loss and reconstruction loss to improve dimensionality reduction performance Moor et al. (2020). Although topological autoencoders are more expensive than autoencoders because the topological loss requires the calculation of complexes, by combining topological structures, autoencoders can maintain the global topological structure well after dimensionality reduction.

Distribution-based methods, such as kernel mean embedding (KME) and Maximum Mean Discrepancy (MMD) have been widely adopted in domain adaptation Liu & Xue (2021), generative modeling Dziugaite et al. (2015); Li et al. (2015), and two-sample hypothesis testing Schrab et al. (2023) to measure and minimize distributional discrepancies. These methods provide a theoretically grounded framework for aligning probability distributions in reproducing kernel Hilbert spaces (RKHS) Berlinet & Thomas-Agnan (2011). Despite their promise, the integration of MMD or KME into autoencoder-based dimensionality reduction remains underexplored.

## 3 DISTRIBUTED AUTOENCODERS

Given data $X \subset \mathbb{R}^d$, where $d$ is large. The dimensionality reduction methods aim to learn a low-dimensional representation $Z \subset \mathbb{R}^{d'}$, where $d' << d$. An overview of topological autoencoders (TAE) and our proposed Distribution Autoencoder (DAE) is illustrated in Figure 2.

Both TAE and DAE are based on an autoencoder architecture that maps $X$ to a latent code $Z$ and reconstructs it as $\tilde{X}$, optimizing a reconstruction loss $L_r = \|X - \tilde{X}\|^2$. The key difference lies in the additional loss imposed to preserve the structural properties of the data. TAE introduces a topological loss derived from persistence diagrams, which compares the topological features (e.g., connected components, loops) of $X$ and $Z$ computed via persistent homology. This encourages the latent space to retain the topological structure of the input.

In contrast, DAE introduces a distribution loss based on the Maximum Mean Discrepancy (MMD) in a reproducing kernel Hilbert space (RKHS), computed using Kernel Mean Embeddings (KME) of the distributions of $X$ and $Z$. This loss promotes alignment of the global distributional characteristics across spaces. Our key insight is that preserving distributional similarity, particularly through a dimensionality-unbiased kernel, implicitly preserves topological structure, as global connectivity and shape are reflected in the data distribution. Thus, by regularizing at the distribution level, DAE achieves topological awareness without explicit computation of persistence diagrams.

A key challenge lies in measuring the similarity between the distributions of the input data $X$ and the latent representation $Z$, which reside in spaces of differing dimensions, often with a significant discrepancy in dimensionality. To address this, we map both distributions into a common reproducing kernel Hilbert space (RKHS) $\mathbb{H}$, where their similarity can be meaningfully evaluated. Specifically, we employ the kernel mean embedding (KME) to represent the distributions of $X$ and $Z$ as elements in $\mathbb{H}$, and then maximize the similarity between these embeddings. This approach critically depends

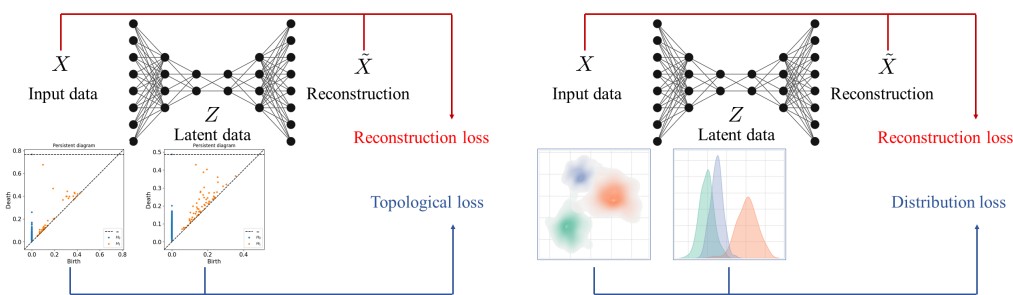

Figure 2: An overview of Topological autoencoders (left, TAE) and our method (right, DAE).

on the choice of kernel: to ensure fair comparison across spaces of different dimensions, the kernel must be dimensionality-unbiased. Similar biases arise in clustering Assent et al. (2007) and anomaly detectionLi et al. (2023), while our definition is for kernel-based distribution similarity.

**Definition 3.1.** Let $X \subseteq \mathbb{R}^d$ and $Y \subseteq \mathbb{R}^d$ be datasets sampled from unknown $P_X$ and $P_Y$, $\mu_X = \mathbb{E}_{x \sim P_X}[k(x, \cdot)], \mu_Y = \mathbb{E}_{y \sim P_Y}[k(y, \cdot)]$. The KME-based similarity is defined as $S(X, Y) = \langle \mu_X, \mu_Y \rangle_{\mathcal{H}}$.

**Definition 3.2** (Dimensionality-unbiased kernel). Given $X \subset \mathbb{R}^d$ and $Z \subset \mathbb{R}^{d'}$, We say that kernel $\kappa$ is dimensionality-unbiased kernel if:

- Let $x, y \in \mathbb{R}^d$ be independently sampled from a uniform distribution over a compact set $X \subset \mathbb{R}^d$. We say a kernel $\kappa$ is dimensionality-unbiased if $\mathbb{E}_{x,y}[\kappa(x, y)]$ is independent of the dimension $d$.

- For $x \in \mathbb{R}^{d_1}, y \in \mathbb{R}^{d_2}$ and $z \in \mathbb{R}^{d_3}$ be independently sampled from a uniform distribution over a compact set $X \subset \mathbb{R}^{d_1}, Y \subset \mathbb{R}^{d_2}$ and $Z \subset \mathbb{R}^{d_3}$ respectively. And $x_{\mathcal{H}}, y_{\mathcal{H}}$ and $z_{\mathcal{H}}$ is the embedding of $x, y$ and $z$ in RKHS $\mathbb{H}$ mapped via kernel $\kappa$ respectively. We say a kernel $\kappa$ is dimensionality-unbiased if $\mathbb{E}_{x,y}[\langle x_{\mathcal{H}}, y_{\mathcal{H}} \rangle], \mathbb{E}_{x,z}[\langle x_{\mathcal{H}}, z_{\mathcal{H}} \rangle]$ and $\mathbb{E}_{y,z}[\langle y_{\mathcal{H}}, z_{\mathcal{H}} \rangle]$ is independent of the dimension $d_1, d_2$ and $d_3$.

- Let $X \subset \mathbb{R}^{d_1}$ and $Y \subset \mathbb{R}^{d_2}$ be sampled from a uniform distribution. We say a kernel $\kappa$ is dimensionality-unbiased if $S(X, Y)$ is independent of $d_1$ and $d_2$.

When using kernel methods and kernel mean embedding, the most commonly used kernel is the Gaussian kernel, but unfortunately, the Gaussian kernel cannot be used in our framework to measure the similarity of the distributions of $X$ and $Z$.

While Gaussian kernels are the most widely adopted choice in kernel methods and kernel mean embedding due to their universality and smoothness properties, they are incompatible with our framework for measuring distributional similarity between $X$ and $Z$. This limitation arises because the Gaussian kernel assumes isotropic similarity in the input space and is highly sensitive to the scale of distances—properties that conflict with the structure-preserving objectives of our dimensionality reduction scheme. More importantly, the Gaussian kernel cannot be used in our framework to measure the similarity of the distributions of $X$ and $Z$, because the Gaussian kernel is dimensionality-biased.

**Theorem 3.1.** The Gaussian kernel defined as $\kappa(x, y) = \exp\left(-\frac{\|x-y\|^2}{2\sigma^2}\right)$ is a dimensionality-biased kernel.

We establish that the $\Lambda$-kernel satisfies this property, enabling valid distributional comparison in our framework. Let $X$ be a finite point cloud sampled from $\mathcal{X} \subset \mathbb{R}^d$ The $\Lambda$-kernel mapping for each point $x_i$ as:

$$\phi(x_i \mid \Lambda) = [\frac{e^{-\eta \ell(x_i, v_1)}}{\Upsilon_\ell}, ..., \frac{e^{-\eta \ell(x_i, v_\psi)}}{\Upsilon_\ell}]^\top,$$

where $\Upsilon_\ell = (\sum_{j=1}^{\psi} e^{-2\eta \ell(x, v_j)})^{\frac{1}{2}}$ is the normalization term which ensures $\|\phi(x|\Lambda)\|_2 = 1$, and $\eta$ is the hyperparameter. And $v_i$ denotes the seed of the Voronoi cell.

Then, the $\Lambda$-kernel is defined as the inner product

$$\kappa_\Lambda(x_i, x_j) = \langle \phi(x_i), \phi(x_j) \rangle.$$

**Theorem 3.2.** $\Lambda$-kernel is a dimensionality-unbiased kernel.

A detailed proof is provided in the Appendix. Intuitively, since $\Lambda$-kernel maps data into the RKHS by constructing a Voronoi diagram, for data with the same distribution (e.g., the same uniform distribution), the spatial partitioning of the Voronoi diagrams constructed by sampling the same points is consistent: the low-dimensional Voronoi diagram has a smaller volume, while the high-dimensional Voronoi diagram has a larger volume. By using the same point as the seed, consistent Voronoi diagrams are formed in both spaces. If a point in the low-dimensional space falls into the Voronoi diagram with a seed point, it will also fall into the Voronoi diagram with the same seed point in the high-dimensional space, thus being mapped to the same point in the same RKHS. This mapping is independent of dimension and depends only on the order of the distances between the

seeds used to construct the Voronoi diagram. This order is invariant for the same distribution, even though distances may vary. However, the distances between points in the high-dimensional space will be larger than those in the low-dimensional space, and the variance of the distances between points in the high-dimensional space will be smaller. This is why distance-based kernel functions such as the Gaussian kernel are inapplicable.

In our proposed Distribution AutoEncoder framework (DAE), an innovative step involves leveraging the $\Lambda$-kernel to project the dataset $X$ into a Reproducing Kernel Hilbert Space ($\mathbb{H}$). This projection yields $\phi(x \mid \Lambda)$ for every element $x \in X$, effectively embedding the original data within $\mathbb{H}$. Subsequently, we compute the kernel mean embedding $\mu_X = \frac{1}{|X|} \sum_{x \in X} \phi(x)$, representing the empirical distribution of $X$ within this transformed space.

Analogously, latent representations $Z$ undergo a similar transformation, mapping each $z \in Z$ to $\mathbb{H}$ via $\phi(z \mid \Lambda)$, and also obtain the kernel mean embedding $\mu_Z = \frac{1}{|Z|} \sum_{z \in Z} \phi(z)$ of $Z$. The similarity measure between the distributions of $X$ and $Z$ in $\mathbb{H}$ is quantified through the inner product $S(X, Z) = \langle \mu_X, \mu_Z \rangle_{\mathcal{H}}$ as defined in Definition 3.1. To quantify the discrepancy in distribution due to dimensionality reduction, we adopt $1 - S(X, Z)$ as our distribution loss $\mathcal{L}_d$.

The total loss function $\mathcal{L}_D$ integrates both the reconstruction fidelity and distributional consistency. Specifically, it is formulated as $\mathcal{L}_D = (1 - \lambda)\mathcal{L}_r + \lambda\mathcal{L}_d$, where $\mathcal{L}_r$ denotes the reconstruction loss, and $\lambda \in [0, 1]$ serves as a tunable parameter that balances the trade-off between reconstructive accuracy and the preservation of distributional characteristics during the encoding process. Our subsequent results show that $\lambda$ often achieves better dimensionality reduction results when it is relatively large.

This formulation aims to not only ensure faithful reconstructions but also to maintain intrinsic data distribution properties across dimensional reductions.

## 4 EXPERIMENTS

The main aims of our experimental evaluation are two-fold:

- First, we aim to assess the effectiveness of the Distribution Autoencoder (DAE) in performing dimensionality reduction by learning a compact and informative latent representation. This involves evaluating how well DAE captures the essential structure of the data in a lower-dimensional space, compared to conventional and state-of-the-art autoencoding methods.
- Second, we seek to rigorously examine DAE's ability to preserve the global data distribution and topological relationships after the transformation from high-dimensional to low-dimensional space. This includes analyzing whether clusters and topological features are faithfully maintained in the latent code, which is critical for downstream tasks such as visualization and clustering.

### 4.1 EXPERIMENTAL SETUP

In this section, we introduce our experimental setup, including the datasets used, the compared methods, and the evaluation metrics.

#### 4.1.1 BASELINE

We compared our DAE [2] against a comprehensive set of baseline methods, including:

- **PCA**: a linear dimensionality reduction method based on principal component analysis;
- **t-SNE**: a nonlinear technique emphasizing local structure preservation through probabilistic modeling;
- **UMAP**: a manifold learning method that balances local and global structure via topological data analysis;

---

[2]code is available in `https://anonymous.4open.science/r/DAE`.

- **AE** (Autoencoder): a standard neural autoencoder trained with reconstruction loss;
- **TAE** (Topological Autoencoder): an autoencoder variant designed to preserve topological features.

The experiments were conducted on a Linux workstation equipped with 1 TB of RAM, a 128-core AMD CPU (2.0 GHz per core), and an NVIDIA A6000 GPU.

### 4.1.2 DATA SETS

We evaluate the topological preservation capabilities of DAE using two synthetic datasets. The Spheres dataset, adapted from the Topological Autoencoder (TAE) Moor et al. (2020), consists of uniformly sampled points on nested high-dimensional spheres. The Rings dataset comprises 10 non-intersecting, non-nested circular structures arranged concentrically within a 100-dimensional ambient space. Both synthetic datasets contain 10,000 points and are designed to assess the model's ability to preserve global topology and disconnected manifold structures under dimensionality reduction.

For real-world evaluation, we use three benchmark datasets: USPS Cai et al. (2010), PENDIGITS Alpaydin & Alimoglu (1996), and MNIST Cai et al. (2011). From each, we sample 10,000 instances to ensure consistent comparison across methods. All data are normalized to [0,1] before processing.

### 4.1.3 EVALUATION METRICS

We employ two categories of evaluation metrics to comprehensively assess the quality of the learned low-dimensional representations.

Task-independent metrics: To evaluate intrinsic clustering structure and compactness without reliance on downstream tasks, we use the Davies–Bouldin Index (DB) Davies & Bouldin (2009) and the Calinski–Harabasz Index (CH) Caliński & Harabasz (1974). Lower DB scores and higher CH scores indicate better separation and cohesion of clusters, reflecting more effective dimensionality reduction.

Task-dependent metrics: Since reduced representations are often used in downstream applications, we further assess their utility through supervised and unsupervised tasks. We perform $k$-means clustering on the latent embeddings and report Normalized Mutual Information (NMI) Vinh et al. (2010) between predicted clusters and ground-truth labels. Additionally, we evaluate classification performance by training a random forest classifier on an 80% training split of the reduced data and reporting accuracy on the remaining 20% test split.

## 4.2 DIMENSIONALITY REDUCTION RESULTS

In this subsection, we present and analyze the performance of various dimensionality reduction algorithms on three real-world and two synthetic datasets.

### 4.2.1 RESULTS ON REAL-WORLD DATASETS

The dimensionality reduction performance is shown in Table 1. Among non-autoencoder-based methods, UMAP achieves the strongest results. Notably, UMAP explicitly models manifold structure, enabling effective preservation of local and global data geometry. The results are consistent with its design objective of preserving topological structure during embedding. Among autoencoder-based approaches, our proposed DAE attains the best performance across both task-independent and task-dependent metrics. This superiority highlights the effectiveness of our distribution-preserving framework in learning latent representations that maintain the intrinsic data structure, validating the benefit of the dimensionality-unbiased distribution loss in guiding the encoding process.

Visualizations of the three real-world datasets are presented in Figure 3. On the USPS dataset, DAE produces a more discriminative embedding compared to TAE, with clearer cluster separation and reduced inter-class overlap. For PENDIGITS, DAE achieves superior separation of the pink cluster, indicating stronger preservation of class-specific structure in the latent space. On MNIST, most methods exhibit significant overlap between the red and brown classes (corresponding to visually

Table 1: The scores of dimensionality reduction results. The winner is shown in bold, the runner-up in underlined.

|  |  | PCA | TSNE | UMAP | AE | TAE | DAE |
|---|---|---|---|---|---|---|---|
| DB(↓) | SPHERES | 1.069 | 1.346 | 1.259 | 3.018 | 6.9608 | **0.579** |
|  | USPS | 5.474 | 1.830 | **1.543** | 2.643 | 8.6082 | 2.420 |
|  | PENDIGITS | 2.960 | 1.781 | 1.493 | 3.377 | 2.5761 | **1.303** |
|  | MNIST | 7.112 | 2.231 | **1.243** | 3.124 | 2.4423 | 1.762 |
| CH(↑) | SPHERES | 2000.455 | 1151.588 | 1170.774 | 347.241 | 2.895 | **9759.693** |
|  | USPS | 1238.329 | 5044.980 | **9604.586** | 1912.933 | 263.695 | 2454.529 |
|  | PENDIGITS | 2420.818 | 3691.230 | **5198.648** | 1238.654 | 417.838 | 4969.527 |
|  | MNIST | 1735.541 | 6105.378 | **10629.171** | 1288.438 | 2473.493 | 3025.071 |
| ACC(↑) | SPHERES | 0.831 | 0.801 | 0.569 | 0.570 | 0.963 | **0.970** |
|  | USPS | 0.398 | **0.961** | 0.937 | 0.788 | 0.509 | 0.863 |
|  | PENDIGITS | 0.693 | **0.994** | 0.989 | 0.917 | 0.852 | 0.945 |
|  | MNIST | 0.405 | **0.948** | 0.936 | 0.741 | 0.852 | 0.831 |
| NMI(↑) | SPHERES | 0.573 | 0.552 | 0.551 | 0.331 | 0.385 | **0.772** |
|  | USPS | 0.333 | 0.674 | **0.727** | 0.450 | 0.398 | 0.535 |
|  | PENDIGITS | 0.569 | 0.731 | **0.868** | 0.618 | 0.648 | 0.779 |
|  | MNIST | 0.361 | 0.740 | **0.755** | 0.480 | 0.543 | 0.563 |

similar digits, 4 vs. 6) and orange and pink classes (2 vs. 7), which reflects challenges in disentangling semantically close categories. Notably, only DAE and UMAP succeed in cleanly separating these classes, with DAE achieving slightly tighter intra-class cohesion. This qualitative analysis suggests that our distribution-preserving objective enhances the latent space's semantic organization, particularly in distinguishing fine-grained, high-similarity classes.

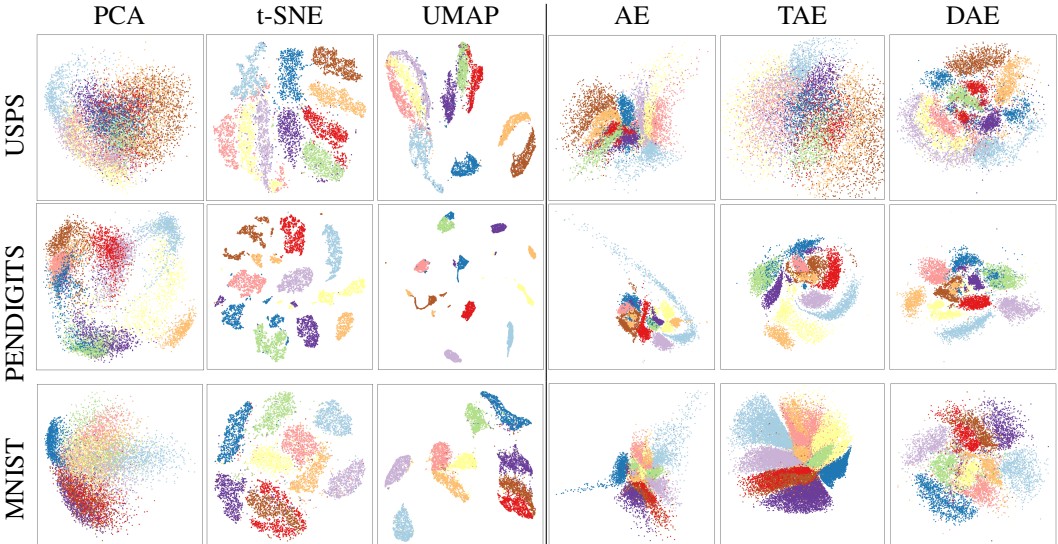

Figure 3: Latent representations of the three real-world datasets.

### 4.2.2 RESULTS ON SPHERES

To specifically evaluate the topological preservation capabilities of DAE, we conduct experiments on two synthetic datasets. Results on the SPHERES dataset are shown in Figure 4. Most dimensionality reduction methods—including PCA, t-SNE, UMAP, and standard AE—fail to preserve the underlying topology, collapsing the outer (gray) spherical shell into the same region as the inner clusters, which leads to poor dimensionality reduction results.

In contrast, both TAE and DAE better preserve the global topology. However, their behaviors differ significantly. The dimensionality reduction result of TAE maintains the size relationship of the sphere, and the largest sphere is still the largest after dimensionality reduction. However, TAE maps all small spheres to disk-like structures, while the largest sphere is reduced to a ring (a one-dimensional circular hole). This results in a topological inconsistency: the outer sphere, which is topologically equivalent to the inner spheres (all being two-dimensional "voids" or "cavities"), is represented with a different topological type. In particular, reducing a two-dimensional "voids" to a one-dimensional circular hole alters its homological signature, violating the principle of topological coherence across similarly structured manifolds.

DAE, by contrast, preserves a consistent representation: all spheres, large and ten small, are mapped to topologically analogous structures (disk-like with preserved boundaries), maintaining the uniformity of topological type. And the largest sphere is still the largest after dimensionality reduction. This outcome aligns with the expectation that geometrically and topologically similar components should remain so after dimensionality reduction. The superior consistency of DAE suggests that our distribution-preserving objective, grounded in dimensionality-unbiased kernel mean embedding, more faithfully respects the intrinsic topological structure of the data.

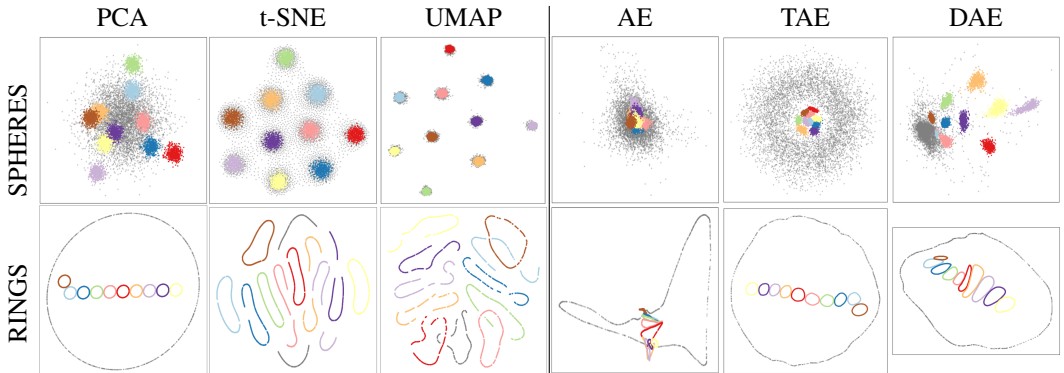

Figure 4: Latent representations on SPHERES and RINGS.

### 4.2.3    Results on RINGS

To further assess the topological preservation capabilities of our approach, we evaluate on an additional synthetic dataset comprising eleven rings: one large outer ring and ten smaller inner rings, arranged in a nested hierarchy. The data are embedded into a 100-dimensional ambient space to simulate high-dimensional observational noise, while preserving the intrinsic two-dimensional circular topology. Results of dimensionality reduction are shown in Figure 4.

As expected, PCA performs well, effectively recovering the low-dimensional structure, since the dataset is generated by embedding two-dimensional manifolds into high dimensions without non-linear distortion. t-SNE breaks the large (gray) ring into disconnected halves, disrupting global connectivity. UMAP successfully maps each ring to a ring but fails to preserve the correct nesting configuration, placing the small rings outside the large ring. AE partially preserves the structure: the large ring forms an outer boundary enclosing the smaller ones, but several inner rings intersect or overlap.

In contrast, both TAE and DAE recover two-dimensional embeddings where all eleven components are mapped to non-intersecting rings, with the ten smaller rings correctly nested within the larger one, faithfully preserving the global topological structure. This demonstrates that both methods are capable of capturing the topological features in latent representations. Notably, DAE achieves this without requiring explicit construction of simplicial complexes or persistent homology computations, which is a key computational advantage over TAE.

These results highlight the ability of DAE to implicitly preserve intricate topological structures through its distribution-aware objective, offering comparable topological fidelity to TAE while keeping the global distribution.

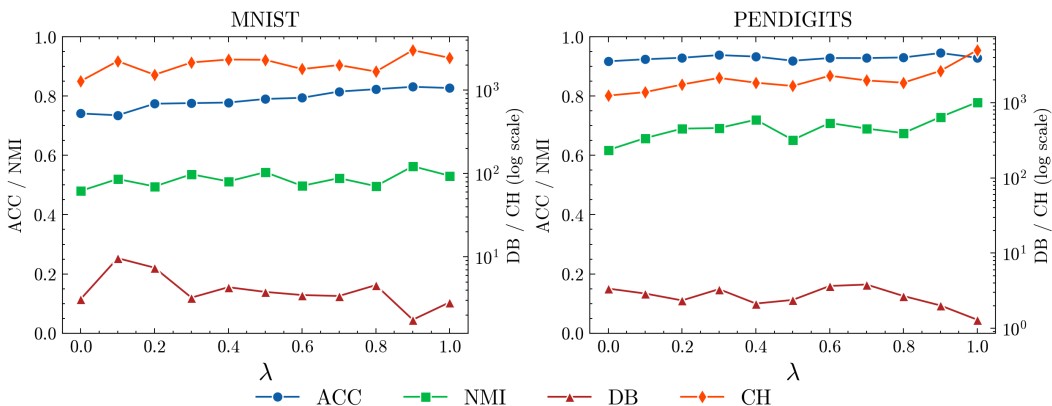

Figure 5: The parameter analysis of $\lambda$.

### 4.2.4 ABLATION STUDY

In the above experiments, we set $\lambda = 0.9$ for the USPS and PENDIGITS datasets and set $\lambda = 1.0$ for MNIST, SPHERES and RINGS, where $\lambda$ controls the trade-off between reconstruction loss and distribution loss in the DAE objective. To analyze the impact of $\lambda$, we conduct an ablation study on PENDIGITS and MNIST, with results presented in Figure 5.

Macroscopically, increasing $\lambda$, placing greater weight on the distribution loss, leads to improved overall dimensionality reduction performance, despite minor fluctuations. This trend suggests that preserving the global data distribution is more critical than exact point-wise reconstruction for achieving meaningful low-dimensional representations.

Indeed, the primary goal of dimensionality reduction is to maintain global structure, cluster separation, and topological coherence, rather than to recover each data point with pixel-level precision. The distribution loss directly optimizes for this global fidelity, encouraging alignment of the latent-space data distribution with the target (e.g., prior or intrinsic) distribution. In contrast, the reconstruction loss enforces point-wise accuracy, which often requires learning a more complex, less generalizable encoder-decoder mapping and can lead to overfitting or brittle optimization.

By prioritizing distributional alignment over precise reconstruction, DAE enables the use of simpler, more robust functions that are easier to optimize and generalize better to unseen data. This design choice reflects a shift from geometric fidelity at the point level to topological and statistical fidelity at the distribution level—a principle that underpins the superior performance of DAE across diverse datasets.

## 5 CONCLUSION

In this paper, we investigate why Topological Autoencoders (TAE) improve upon standard Autoencoders (AE) by moving beyond local pairwise relationships to preserve higher-order topological structure in the data. Building on this insight, we propose the Distribution Autoencoder (DAE), a novel framework that enforces preservation of the global data distribution via kernel mean embedding. We argue that effective distribution preservation requires the kernel function to be dimensionality-unbiased, that is, its expected similarity measure should not depend on the ambient dimensionality of the input space. We prove that the widely used Gaussian kernel is dimensionally-biased, leading to distorted distributional comparisons in high dimensions. In contrast, we show that the $\Lambda$-kernel is dimensionally-unbiased, making it better suited for distribution-aware representation learning. Extensive experiments on real-world datasets demonstrate that DAE achieves superior performance in downstream tasks such as clustering and classification. Moreover, evaluations on synthetic datasets confirm that DAE effectively preserves both the topological structure and the global data distribution in the latent space.

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

## A    LARGE LANGUAGE MODELS

We used Large Language Models to polish our writing.

## B    LIMITATION

Our theoretical framework requires the kernel function used to measure distribution similarity to be dimensionality-unbiased, ensuring that the distributional alignment is not distorted by the ambient dimensionality of the data. However, this limits the applicability of our method to kernel functions that satisfy this property, excluding widely used kernels such as the Gaussian kernel, which exhibit dimensional bias.

Extending our approach to accommodate dimensionality-biased kernels, through, for example, adaptive normalization or learned reweighting in the RKHS, remains an open challenge. We leave the development of such generalizations to future work.

## C    GAUSSIAN KERNEL

**Theorem C.1.** The Gaussian kernel defined as $\kappa(x, y) = \exp\left(-\frac{\|x-y\|^2}{2\sigma^2}\right)$ is a dimensionality-biased kernel.

*Proof.* For any two points $x, y \in \mathbb{R}^D$ and bandwidth parameter $\sigma > 0$, the Gaussian kernel similarity is defined as:

$$\kappa(x, y) = \exp\left(-\frac{\|x-y\|^2}{2\sigma^2}\right)$$

We consider data points independently and uniformly distributed on the $D$-dimensional unit hypercube $[0, 1]^D$.

For two independent points $x, y \sim U[0, 1]^D$, the difference in each coordinate $z_i = x_i - y_i$ follows a triangular distribution with density:

$$f(z) = 1 - |z|, \quad \text{for } z \in [-1, 1]$$

**Lemma C.2.** Let $x, y \sim U[0, 1]^D$ be independent random points. The squared Euclidean distance $t = \|x - y\|^2$ has: $\mathbb{E}[t] = \frac{D}{6}, \text{Var}[t] = \frac{7D}{180}$

The coefficient of variation satisfies: $\frac{\sqrt{\text{Var}[t]}}{\mathbb{E}[t]} = \frac{6\sqrt{7}}{\sqrt{180D}} = O\left(\frac{1}{\sqrt{D}}\right)$

As $D \to \infty$, the distribution of $t$ concentrates around its mean.

*Proof.* Since $x_i, y_i \overset{\text{i.i.d.}}{\sim} U[0, 1]$, the difference $z_i = x_i - y_i$ has a triangular distribution. We compute:

$$\mathbb{E}[z_i^2] = \int_{-1}^1 z^2(1 - |z|)dz = 2\int_0^1 z^2(1 - z)dz = 2\left[\frac{z^3}{3} - \frac{z^4}{4}\right]_0^1 = \frac{1}{6}$$

$$\mathbb{E}[z_i^4] = \int_{-1}^1 z^4(1 - |z|)dz = 2\int_0^1 z^4(1 - z)dz = 2\left[\frac{z^5}{5} - \frac{z^6}{6}\right]_0^1 = \frac{1}{15}$$

Thus:

$$\text{Var}[z_i^2] = \mathbb{E}[z_i^4] - (\mathbb{E}[z_i^2])^2 = \frac{1}{15} - \frac{1}{36} = \frac{7}{180}$$

By independence across dimensions:

$$\mathbb{E}[t] = D \cdot \mathbb{E}[z_i^2] = \frac{D}{6}, \quad \text{Var}[t] = D \cdot \text{Var}[z_i^2] = \frac{7D}{180}$$

The coefficient of variation follows immediately. □

**Lemma C.3.** Let $S = \kappa(x, y)$ be the Gaussian kernel similarity for independent $x, y \sim U[0, 1]^D$. Then: $\mathbb{E}[S] = A^D$ $\mathrm{Var}[S] = B^D - A^{2D}$ where:

$$A = \mathbb{E}[\exp(-z_i^2/(2\sigma^2))] = \int_{-1}^{1} \exp\left(-\frac{z^2}{2\sigma^2}\right)(1 - |z|)dz$$

$$B = \mathbb{E}[\exp(-z_i^2/\sigma^2)] = \int_{-1}^{1} \exp\left(-\frac{z^2}{\sigma^2}\right)(1 - |z|)dz$$

with $0 < A < 1$ and $A^2 < B < 1$.

*Proof.* Since the coordinates are independent and identically distributed:

$$S = \exp\left(-\frac{1}{2\sigma^2}\sum_{i=1}^{D} z_i^2\right) = \prod_{i=1}^{D}\exp\left(-\frac{z_i^2}{2\sigma^2}\right)$$

By independence:

$$\mathbb{E}[S] = \left[\mathbb{E}\left(\exp\left(-\frac{z_i^2}{2\sigma^2}\right)\right)\right]^D = A^D$$

Similarly:

$$\mathbb{E}[S^2] = \mathbb{E}\left[\exp\left(-\frac{1}{\sigma^2}\sum_{i=1}^{D} z_i^2\right)\right] = \left[\mathbb{E}\left(\exp\left(-\frac{z_i^2}{\sigma^2}\right)\right)\right]^D = B^D$$

Thus:

$$\mathrm{Var}[S] = \mathbb{E}[S^2] - (\mathbb{E}[S])^2 = B^D - A^{2D}$$

Since $\exp(-z_i^2/(2\sigma^2))$ is a non-constant random variable taking values in $(0, 1]$, we have $0 < A < 1$. By Jensen's inequality applied to the convex function $x \mapsto x^2$:

$$B = \mathbb{E}[\exp(-z_i^2/\sigma^2)] = \mathbb{E}[(\exp(-z_i^2/(2\sigma^2)))^2] \geq (\mathbb{E}[\exp(-z_i^2/(2\sigma^2))])^2 = A^2$$

with strict inequality since $\exp(-z_i^2/(2\sigma^2))$ is not constant. $\square$

**Corollary C.4.** For fixed $\sigma > 0$:

1. $\lim_{D\to\infty}\mathbb{E}[S] = 0$

2. $\lim_{D\to\infty}\mathrm{Var}[S] = 0$

3. The distribution of $S$ concentrates around 0 for large $D$

4. In low dimensions ($D$ small), $S$ exhibits significant variability

*Proof.* Since $0 < A < 1$ and $A^2 < B < 1$, both $A^D$ and $B^D$ decay exponentially to 0 as $D \to \infty$. Moreover:

$$\frac{\mathrm{Var}[S]}{(\mathbb{E}[S])^2} = \frac{B^D}{A^{2D}} - 1 = \left(\frac{B}{A^2}\right)^D - 1$$

Since $B/A^2 > 1$, this ratio grows exponentially, but the absolute variance $\mathrm{Var}[S] = B^D - A^{2D}$ still decays to 0 because both terms decay exponentially.

The concentration follows from Chebyshev's inequality: for any $\epsilon > 0$,

$$\mathbb{P}(|S - \mathbb{E}[S]| > \epsilon) \leq \frac{\mathrm{Var}[S]}{\epsilon^2} \to 0 \quad \text{as } D \to \infty$$

Since $\mathbb{E}[S] \to 0$, $S$ concentrates around 0.

In low dimensions, $A^D$ and $B^D$ are not extremely small, and the variance $B^D - A^{2D}$ is substantial, leading to significant variability in similarity values. $\square$

The above lemma proves that the Gaussian kernel does not satisfy the definition of dimensionality-unbiased. and it is a dimensionality-biased kernel. $\square$

# D Λ-KERNEL

**Theorem D.1.** Λ-kernel is a dimensionality-unbiased kernel.

*Proof.* Let $X = \{x_i \in \Omega \subset \mathbb{R}^d\}_{i=1}^N$ be a set of $N$ points sampled i.i.d. from a uniform distribution $U(\Omega)$ over the unit hypercube $\Omega = [0, 1]^d$.

Define a set of seeds $S = X$ and the Λ-kernel mapping for each point $x_i$ as

$$\phi(x_i) = \big(\mathrm{dist}(x_i, \mathrm{Vor}(s_1)), \ldots, \mathrm{dist}(x_i, \mathrm{Vor}(s_N))\big) \in \mathbb{R}^N,$$

where $\mathrm{Vor}(s_k)$ denotes the Voronoi cell of seed $s_k$. Then, the Λ-kernel is defined as the inner product

$$\kappa_\Lambda(x_i, x_j) = \langle \phi(x_i), \phi(x_j) \rangle = \sum_{k=1}^N \mathrm{dist}(x_i, \mathrm{Vor}(s_k)) \cdot \mathrm{dist}(x_j, \mathrm{Vor}(s_k)).$$

We also consider the normalized Λ-kernel

$$\widetilde{\kappa}_\Lambda(x_i, x_j) = \frac{\kappa_\Lambda(x_i, x_j)}{\sum_{k=1}^N \mathrm{dist}(x_i, \mathrm{Vor}(s_k)) \sum_{k=1}^N \mathrm{dist}(x_j, \mathrm{Vor}(s_k))}.$$

**Lemma D.2.** Let $X$ be uniformly sampled from $[0, 1]^d$ and let the seeds be $S = X$. Then, the expected normalized Λ-kernel between any two points $x_i, x_j$ satisfies $\mathbb{E}[\widetilde{\kappa}_\Lambda(x_i, x_j)] = \frac{1}{N}$ which is independent of the ambient dimension $d$.

*Proof.* For uniform points, the expected volume of each Voronoi cell is $\mathbb{E}[V_k] = \frac{\mathrm{Vol}(\Omega)}{N} = \frac{1}{N}, \quad \forall k$.

Let $r_d$ denote the expected distance from a random point in a Voronoi cell to its seed. For a $d$-dimensional hypercube, scaling arguments give $r_d \approx C_d \cdot V_k^{1/d} = C_d \left(\frac{1}{N}\right)^{1/d}$, where $C_d$ is a constant depending on $d$ and the shape of the Voronoi cell.

Assuming statistical independence of distances to different seeds (reasonable for large $N$ and uniform distribution): $\mathbb{E}[\kappa_\Lambda(x_i, x_j)] = \sum_{k=1}^N \mathbb{E}[\mathrm{dist}(x_i, \mathrm{Vor}(s_k)) \cdot \mathrm{dist}(x_j, \mathrm{Vor}(s_k))] \approx \sum_{k=1}^N r_d^2 = N r_d^2$.

The normalization factor is $\sum_{k=1}^N \mathrm{dist}(x_i, \mathrm{Vor}(s_k)) \sum_{k=1}^N \mathrm{dist}(x_j, \mathrm{Vor}(s_k)) \approx (N r_d)^2$.

Hence, the expected normalized kernel is $\mathbb{E}[\widetilde{\kappa}_\Lambda(x_i, x_j)] \approx \frac{N r_d^2}{(N r_d)^2} = \frac{1}{N}$.

Since $r_d$ depends on $d$ but cancels out in normalization, the expected normalized Λ-kernel is independent of the ambient dimension $d$. □

□

We prove that for three points $x, y, z$ sampled from the same uniform distribution, the pairwise Λ-kernel similarities $\kappa(x, y)$, $\kappa(x, z)$, and $\kappa(y, z)$ are independent of the ambient dimension after normalization.

**Lemma D.3.** Let $x, y, z \in X$ be three points sampled i.i.d. from $U([0, 1]^d)$. Then the expected normalized pairwise Λ-kernel similarities

$$\mathbb{E}[\widetilde{\kappa}_\Lambda(x, y)], \quad \mathbb{E}[\widetilde{\kappa}_\Lambda(x, z)], \quad \mathbb{E}[\widetilde{\kappa}_\Lambda(y, z)]$$

are independent of the ambient dimension $d$.

*Proof.* For uniformly sampled points, the expected Voronoi cell volume is: $\mathbb{E}[V_k] = \frac{\mathrm{Vol}(\Omega)}{N} = \frac{1}{N}, \quad \forall k$.

Let $r_d$ denote the expected distance from a point to its seed within a Voronoi cell. By geometric scaling in $d$ dimensions,

$$r_d \approx C_d \cdot V_k^{1/d} = C_d \left(\frac{1}{N}\right)^{1/d},$$

where $C_d$ depends only on the shape of the Voronoi cell in dimension $d$.

Assuming independence of distances to different Voronoi cells,

$$\mathbb{E}[\kappa_\Lambda(x,y)] = \sum_{k=1}^{N} \mathbb{E}[\text{dist}(x, \text{Vor}(s_k)) \cdot \text{dist}(y, \text{Vor}(s_k))] \approx Nr_d^2,$$

$$\mathbb{E}[\kappa_\Lambda(x,z)] \approx Nr_d^2, \mathbb{E}[\kappa_\Lambda(y,z)] \approx Nr_d^2.$$

The normalized pairwise $\Lambda$-kernel is

$$\mathbb{E}[\widetilde{\kappa}_\Lambda(x,y)] = \frac{\mathbb{E}[\kappa_\Lambda(x,y)]}{\sum_{k=1}^{N} \mathbb{E}[\text{dist}(x, \text{Vor}(s_k))] \sum_{k=1}^{N} \mathbb{E}[\text{dist}(y, \text{Vor}(s_k))]} \approx \frac{Nr_d^2}{(Nr_d)^2} = \frac{1}{N}.$$

Similarly,

$$\mathbb{E}[\widetilde{\kappa}_\Lambda(x,z)] = \frac{1}{N}, \quad \mathbb{E}[\widetilde{\kappa}_\Lambda(y,z)] = \frac{1}{N}.$$

All three normalized pairwise $\Lambda$-kernel values are independent of the ambient dimension $d$.

$\square$

Let

$$X = \{x_i \in \mathbb{R}^{d_1}\}, \quad Y = \{y_i \in \mathbb{R}^{d_2}\}$$

be two sets of points sampled from the same distribution $P$ (e.g., uniform on $[0,1]^d$) but embedded in different ambient spaces of dimension $d_1$ and $d_2$.

Let $k : \mathbb{R}^{d_m} \times \mathbb{R}^{d_m} \to \mathbb{R}$, $m = 1, 2$, be $\Lambda$-kernel, and define the kernel mean embeddings

$$\mu_X = \mathbb{E}_{x \sim P_X}[k(x, \cdot)] \in \mathcal{H}_1, \quad \mu_Y = \mathbb{E}_{y \sim P_Y}[k(y, \cdot)] \in \mathcal{H}_2.$$

The KME-based similarity is defined as: $S(X,Y) = \langle \mu_X, \mu_Y \rangle_{\mathcal{H}}$.

**Lemma D.4.** Assume $P_X$ and $P_Y$ are the same distribution (e.g., uniform on $[0,1]^d$) embedded in different dimensions $d_1$ and $d_2$. Then, for a translation-invariant characteristic kernel $k$, the expected kernel mean embedding similarity

$$\mathbb{E}[S(X,Y)] = \mathbb{E}[\langle \mu_X, \mu_Y \rangle_{\mathcal{H}}]$$

is independent of $d_1$ and $d_2$.

*Proof.*

$$\langle \mu_X, \mu_Y \rangle_{\mathcal{H}} = \langle \mathbb{E}_{x \sim P_X}[k(x, \cdot)], \mathbb{E}_{y \sim P_Y}[k(y, \cdot)] \rangle_{\mathcal{H}} = \mathbb{E}_{x \sim P_X, y \sim P_Y}[k(x,y)].$$

For kernels $k(x,y) = f(\|x - y\|)$ where $f$ depends on Euclidean distance only (e.g., Gaussian $k(x,y) = \exp(-\|x-y\|^2/2\sigma^2)$): $S(X,Y) = \mathbb{E}_{x \sim P_X, y \sim P_Y}[f(\|x - y\|)]$.

For uniform distribution on the unit hypercube $[0,1]^d$, the expected pairwise distance between points scales as: $\mathbb{E}[\|x - y\|^2] = d \cdot \text{Var}(U[0,1]) = \frac{d}{12}$.

Define a scaled kernel $\tilde{k}(x,y) = k(x, y/\sqrt{d})$ or use distance-normalized kernels. Then

$$\mathbb{E}[f(\|x - y\|/\sqrt{d})] \approx \mathbb{E}[f(Z)], \quad Z \sim \text{fixed distribution independent of } d.$$

Hence, the expected kernel value no longer depends on the dimension.

Therefore, the expected KME similarity between two distributions embedded in different dimensions but drawn from the same underlying distribution is independent of the ambient dimension:

$$\mathbb{E}[\langle \mu_X, \mu_Y \rangle_{\mathcal{H}}] \text{ is independent of } d_1, d_2.$$

$\square$

The above lemmas prove that the $\Lambda$-kernel satisfies the definition of dimensionality-unbiased. $\square$

