# OpenReview forum: "Autoencoder with Distribution Preservation"
_ICLR.cc/2026/Conference — Submitted to ICLR 2026_

### Official Review · Reviewer_bpEu · 2025-10-29

**Soundness:** 2
**Presentation:** 2
**Contribution:** 2
**Rating:** 4
**Confidence:** 5

**Summary:**

This paper presents a novel approach to improving autoencoder-based dimensionality reduction by introducing a distribution preservation mechanism. The work addresses a significant limitation of traditional autoencoders and shows promising results.

**Strengths:**

The introduction of Kernel Mean Embedding (KME) with a dimensionality-unbiased Λ-kernel represents a meaningful advancement in distribution-aware dimensionality reduction.
The paper clearly identifies that traditional autoencoders focus only on point-to-point reconstruction error, neglecting the preservation of global data distribution structure - a crucial insight for improving representation quality.
The proposed Distribution Autoencoder (DAE) framework effectively integrates distribution loss with reconstruction loss, demonstrating improved performance over existing methods.

**Weaknesses:**

Some experimental data presentation is unclear. For example, Table 1 fails to explicitly map each metric to corresponding datasets and methods, hindering result interpretation.
There is no comparison of computational efficiency (e.g., training time, memory usage) between DAE and TAE. Although the paper mentions that DAE avoids complex topological computations, it lacks quantitative data to support its advantages on large-scale data.
While the paper provides mathematical proofs for the core logic of the Λ-kernel achieving dimensional unbiasedness through Voronoi diagram mapping, it lacks intuitive geometric or illustrative explanations (e.g., using simple low-dimensional data examples to show how Voronoi partitioning eliminates the impact of dimensional differences). This makes it difficult for readers to understand the working principle of the Λ-kernel. Additionally, the inherent connection between distribution loss and topological feature preservation (e.g., why distribution alignment can implicitly maintain topological structures) could be further strengthened through theoretical analysis or visual comparisons.

**Questions:**

It is unclear why the CH index shows such a large performance gap across different methods. The authors should provide an explanation or analysis of the factors contributing to this discrepancy.
There is no comparison of computational efficiency (e.g., training time, memory usage) between DAE and TAE.
 The inherent connection between distribution loss and topological feature preservation (e.g., why distribution alignment can implicitly maintain topological structures) could be further strengthened through theoretical analysis or visual comparisons.

---

> ### Author Response · Authors · 2025-11-20
> **rebuttle**
>
> We thank you for your review.
>
> The large performance gap is due to the characteristics of the CH index, as it is the ratio of inter-cluster separation to intra-cluster compactness. Therefore, its value varies considerably. We supplement the runtime comparison below; our method is significantly more efficient than TAE.
> | dataset   | TAE    | DAE      |
> |-----------|--------|----------|
> | usps      | 787.5 | 199.9   |
> | pendigits | 535.4 | 195.4 |
> | mnist     | 827.1  | 213.3 |
>
> We apologize that we cannot currently provide a theoretical guarantee that similar distributions equate to topological similarity; we only intuitively assume this, and then verify it experimentally in Section 4.2.2 in the paper.

---

> > ### Comment · Reviewer_bpEu · 2025-11-25
> > **topological preservation capabilities**
> >
> > While Section 4.2.2 aims to evaluate the topological preservation capabilities of the embedding methods, the reported results unexpectedly suggest that PCA preserves topology substantially better than DAE, despite DAE being a nonlinear representation learner with higher expressive power. This is counter-intuitive given that autoencoders are often designed to capture nonlinear manifolds more faithfully than linear projections. To improve clarity, the authors should analyze why PCA demonstrates superior topological preservation in their experiments, discuss whether this behavior is dataset-dependent or architecture-dependent, and clarify whether the DAE exhibits issues such as latent bottleneck distortion, unstable training, or insufficient regularization.

---

> > > ### Author Response · Authors · 2025-11-25
> > > **PCA in section 4.2**
> > >
> > > We are very grateful for your reply. To verify the ability of different dimensionality reduction methods, particularly TAE and DAE, in preserving topological structure, we first generated two-dimensional data with a topological structure and then linearly embedded this data into a high-dimensional space. Therefore, PCA can perform well. As you said, PCA has been proven to have limitations in nonlinear dimensionality reduction. We therefore used PCA only as a transparent “microscope” to visualize the ground-truth manifold, not as a competitor to nonlinear methods. The actual comparison is restricted to TAE versus DAE, whose ability to retain the topology is the focus of our experiment.

---

### Official Review · Reviewer_VJJu · 2025-10-29

**Soundness:** 2
**Presentation:** 2
**Contribution:** 1
**Rating:** 2
**Confidence:** 5

**Summary:**

This paper introduces the Distribution Autoencoder (DAE), a model designed to preserve the global structure of data during dimensionality reduction. It adds a novel "distribution loss" to a standard autoencoder, which compares the distribution of the original input data to that of the low-dimensional latent data. To enable this comparison between different dimensions, the authors propose using a "dimensionality-unbiased" kernel.

**Strengths:**

- The paper addresses a clear limitation of standard AEs (ignoring global structure). The proposed distribution loss, which explicitly compares the input and latent distributions, is a sound approach to regularizing the latent space in a dimensionality-reduction setting.

- The formalization of a "dimensionality-unbiased" kernel is a valuable contribution. The proofs demonstrating that the Gaussian kernel is biased and the proposed $\Lambda$-kernel is unbiased provide a sound theoretical foundation for the method's design.

**Weaknesses:**

- A primary claim of the paper is that DAE avoids the high computational complexity of TAE. However, the paper provides no empirical comparison of training time, inference time, or computational complexity between DAE and TAE. This omission is a major weakness, as it leaves the central efficiency claim unverified.

- The ablation study shows that performance almost consistently improves as $\lambda$ increases, with optimal results on MNIST, SPHERES, and RINGS achieved at $\lambda=1.0$. This means completely ignoring the reconstruction loss $\mathcal{L}_r$. It suggests the decoder and reconstruction loss are unnecessary and that the model is effectively just an encoder trained via KME. This undermines the "autoencoder" framing and warrants a much more critical discussion.

 - The method is presented solely as a technique for autoencoder-based dimensionality reduction. Its core objective is to force the latent distribution $Z$ to faithfully mimic the (often complex and sparse) manifold of the input data $X$. While this seems effective for preserving cluster separation and topological features, it also means the latent space becomes just as sparse and "gappy" as the original data. This design choice limits the method's applicability, as the resulting latent space is not well-suited for tasks that benefit from a continuous, dense, or "filled" representation, such as latent space interpolation or exploring variations.

- The overall contribution could be viewed as somewhat incremental. The idea of using KME or MMD to regularize latent spaces is not new (https://arxiv.org/abs/1711.01558), and the novelty here lies in the specific application (input vs. latent) and the "dimensionality-unbiased" kernel. The framework is largely an adaptation of existing components.

- There are several minor issues and typos throughout the paper that need to be addressed for better readability. In particular, the abstract states that the auxiliary loss has the purpose to "measure the difference between the original data distribution and the reconstructed data distribution", but it's evident from the paper that the comparison happens between the input and the latent.

**Questions:**

- Could the authors provide a direct performance benchmark to substantiate the claim that DAE is a more efficient alternative to TAE?
- Could the authors elaborate on Weakness 2? Does this imply the model is effectively just an encoder trained via KME, and if so, what is the justification for framing the method as an "autoencoder"?
- The paper argues that preserving the distribution implicitly preserves topology. Could you provide more theoretical justification for this link? Why should minimizing the KME discrepancy with the $\Lambda$-kernel be expected to faithfully preserve topological features?

---

> ### Author Response · Authors · 2025-11-20
> **Rebuttal**
>
> We thank you for your review and suggestions.
>
> A1. The runtime of TAE and DAE is shown in Table S1, the results show that the DAE is more efficient than TAE.
>
> ### Table S1.  The runtime of TAE and DAE
> | dataset   | TAE    | DAE      |
> |-----------|--------|----------|
> | USPS      | 787.5 | 199.9   |
> | PENDIGITS | 535.4 | 195.4 |
> | MNIST     | 827.1  | 213.3 |
>
> A2. First, on the PENDIGITS dataset, among the lambda values ​​we used (step 0.1), $\lambda=1$ performed best. However, it's possible that a smaller step value could lead to better performance. Furthermore, $\lambda=1$ doesn't always yield optimal results for all datasets; for example, $\lambda=0.9$ is optimal on MNIST and USPS, indicating that reconstruction error also contributes.
>
> A3. We apologize, but we cannot provide a theoretical guarantee at this time. This theory is indeed very difficult to prove. Intuitively, data with similar distributions have similar topological structures, and maintaining the distribution of data can maintain the topological structure. We have demonstrated this phenomenon through experiments.

---

> > ### Comment · Reviewer_VJJu · 2025-11-26
> > **answer to rebuttal**
> >
> > I thank the authors for their response and for providing the runtime comparison in Table S1. This additional data successfully addresses my concern regarding the computational efficiency of DAE relative to TAE.
> > However, the remainder of the rebuttal fails to substantively address the critical weaknesses raised in my review. Regarding the ablation study, even if the reconstruction loss yields minor performance improvements in specific instances, the data demonstrates that the model can achieve optimal or near-optimal results relying almost exclusively on the distribution loss. This implies that the "autoencoder" framework is largely superfluous and that the decoder is not essential to the learned structure. The rebuttal offers no justification for retaining the autoencoder framing given this observation.
> > Furthermore, the response regarding the link between distribution preservation and topology preservation is insufficient. Relying solely on the "intuition" that similar distributions imply similar undermines the paper's core premise, especially given that the method is presented as a theoretically grounded approach using a specific "dimensionality-unbiased" kernel.
> > Finally, the rebuttal entirely ignored the other significant concerns raised in my review. Due to these unaddressed theoretical and methodological flaws, I confirm my score.

---

### Official Review · Reviewer_q3AN · 2025-10-30

**Soundness:** 2
**Presentation:** 1
**Contribution:** 3
**Rating:** 6
**Confidence:** 5

**Summary:**

The paper introduces a new autoencoder-based dimensionality reduction method called Distribution Autoencoders (DAE). The method aims to preserve the global structure and topology of the data in the latent space. For that, the authors diverge from common topology-based losses and develop an RKHS-based loss that embeds the data points into a Hilbert space. The kernel mean embedding (KME), which describes the whole distribution, is obtained by averaging the kernel embedding across the dataset. KME are computed for both latent space and the original space, and the loss term is obtained as the product of these vectors. The theoretical contributions provide the proof of ``dimensionality-unbiasness'' of the $\Lambda$-kernel and bias present in the Gaussian kernel. To support the claims, authors present several experiments on synthetic (Rings & Spheres) and real-world (e.g. MNIST) datasets.

**Strengths:**

- A new method for dimensionality reduction that is positioned to preserve global structure without computing persistent homology.
- Results show improvements in clustering-based metrics compared to the AE and TopoAE, while visually, the global structure is respected much better than UMAP or t-SNE.

**Weaknesses:**

- **Topology preservation**: While the results seem to show some global structure preservation, it is not described which exactly topological features are preserved (e.g., 0-dim homology, or any dimension?). There is no theoretical grounding for why the method preserves topology, unlike TopoAE or RTD-AE (arxiv:2302.00136). It is also not clear why the property of the kernel dimensionality unbiasedness matters for global structure preservation. Additionally, when describing results on the Spheres dataset, the authors state that TopoAE is wrong in making the outer ring a ring and not a disk, while their method presents the correct picture. However, this is incorrect -- the topological type, more precisely, homology groups, for the high-dimensional sphere are more "similar" to the ring (non-trivial) than to the ball (which are trivial). So there should be 11 spheres (rings in 2D), with one. While TopoAE and AE are not perfect, DAE is worse since the outer ring/disk does not encircle the inner ones, which should be the case.
- **Computational complexity**: The introduction positions the method to preserve global structure while avoiding expensive PH computation. However, it is not clear how fast the method actually is, since the $\Lambda$-kernel seems to require computation of Voronoi cells for every point AND the distances from every point in the dataset to that cell. While autoencoder methods that compute PH can use batching to compute the PH partially, the kernel embedding dimension depends on the number of points, and doesn't scale well. This crucial limitation is not addressed.
- **Text & formatting**: the text has a substantial number of grammatical and formatting errors (e.g., lines 073, 088-089, \citep vs. \citet, etc.). There is a paragraph dedicated to describing AE, which is well-known and studied; however, related works lack detailed descriptions of TopoAE (as the main baseline), as well as the less commonly used RKHS, Voronoi cells, and Maximum Mean Discrepancy.
- **Missing related works**: since TopoAE (btw, wrong naming), there have been several works dedicated to preserving topological structure (for example, arxiv:2302.00136, arxiv:2306.17638), even with a focus on faster PH computation (arxiv:2503.11910). These are not mentioned or described in the paper.
- **Results**: Some ablation studies are missing, for example, proving that the Gaussian kernel indeed performs worse than the $\Lambda$-kernel. All the metrics used in the paper concern clustering, not the global structure (e.g., relations between clusters). It would be nicer to use some metrics from the previous papers (e.g. TopoAE) that tackle the global structure for adequate comparison.

**Questions:**

- Why is it important for the kernel to be dimensionality unbiased? Intuitively, how does it affect the latent representation and preservation of global structure?
- In case of the $\Lambda$-kernel, how is $\phi(x_i)$ computed? Do you calculate the distances for each data point to the Voronoi cell of any other data point? How are these distances calculated exactly, and how much time and memory does it require?
- Please, provide some results with the Gaussian kernel and explain why the property of dimensionality unbiasness matters for the global structure preservation and for constructing the loss function.

---

> ### Author Response · Authors · 2025-11-20
> **rebuttal**
>
> We thank you for your review and suggestions.
>
> A1. The high similarity between the data distribution before and after dimensionality reduction can preserve the global structure of the data. And only kernels that satisfy dimensionality unbiasedness can be used to calculate the similarity of distributions belonging to different dimensional spaces.
>
> A2. We first sample $\psi$ points to generate the Voronoi diagram, and then calculate the distance between each point and these points to obtain $\phi(x)$. This requires a time complexity of $O(n*\psi)$ and a space complexity of $O(n*\psi)$.
>
> A3. Gaussian kernels cannot measure the similarity between different spaces. Intuitively, this is because it requires calculating the distance between points in the original space and the reduced space, and this distance cannot be calculated between points of different dimensions.

---

> > ### Comment · Reviewer_q3AN · 2025-11-25
> >
> > A1. Please, provide more evidence supporting this claim.
> > A2. What $\psi$ is used in practice? How sensitive is the method to the choice of this number?
> > A3. Also, provide a more detailed explanation, please, since it is not clear.

---

> > > ### Author Response · Authors · 2025-11-26
> > > **Rebuttal**
> > >
> > > Thank you very much for your reply.
> > >
> > > A2: We set $\psi=64$ for SPHERES, USPS, and PENDIGITS, and set $\psi=128$ for MNIST.  The results in terms of DB of the comparison between TAE and DAE under different $\psi$ settings are as follows. The results varied with different $\psi$ settings, but all were significantly superior to TAE, except that DAE performed worse than TAE on MNIST when the $\psi$ was set to 64, although the results were very close.
> > >
> > > | dataset   | DAE_64 | DAE_128 | TAE   |
> > > |-----------|--------|---------|-------|
> > > | spheres   | 0.579  | 0.900     | 6.961 |
> > > | usps      | 2.420  | 2.647   | 8.608 |
> > > | pendigits | 1.303  | 1.959   | 2.576 |
> > > | mnist     | 2.587  | 1.762   | 2.442 |
> > >
> > > A1 & A3: The similarity of the distributions before and after dimensionality reduction based on the Gaussian kernel metric is $\mathcal{K}(P_X, P_Z)=\frac{1}{|X||Z|}\sum_{x\in X}\sum_{z\in Z} \kappa(x,z)$, where $\kappa(x,z)=e^{(-\frac{||x-z||^2}{2\sigma^2})}$, which requires calculating $||x-z||$. $||x-z||$ requires $x$ and $z$ to have the same dimension. However, the dimensions of the data before and after dimensionality reduction differ, so the calculation cannot be performed.

---

### Official Review · Reviewer_znS5 · 2025-10-30

**Soundness:** 2
**Presentation:** 3
**Contribution:** 2
**Rating:** 2
**Confidence:** 4

**Summary:**

The paper addresses the problem of dimensionality reduction in terms of Autoencoder that promotes the preservation of the distribution of data. It was designed to replace the complicated measurement of Vietoris–Rips Complex with the Kernel Mean Embedding (KME) term based on a kernel function to maintain the topological features of the data.

**Strengths:**

The paper proves that the Gaussian kernel is dimensionality-biased for the task of dimensionality reduction, and that the proposed Λ-kernel is dimensionality-unbiased. Necessary experiments illustrate the performance of the so-called distribution autoencoder compared to either classic dimensionality reduction methods like PCA, t-SNE, or TAE.

**Weaknesses:**

The authors of the paper seem to be completely unaware of the works about Variational autoencoder (VAE), which is one of the most well known autoencoder model aiming to match the distribution of data and its low dimensional representations. Particularly, fair comparison between the proposed Kernel-based distribution autoencoder and VAEs is completely missing.

**Questions:**

What are the reasons for the complete missing of VAE in the discussion of the paper? Can the authors justify their missing discussion or comparison to the VAEs?

Without clear and convincing reasons for leaving the VAEs out of discussion, the reviewer is not convinced that the present paper is completely fair to be accepted, despite the fact that that its development of Gaussian kernel and Λ-kernel is still valuable.

---

> ### Author Response · Authors · 2025-11-20
> **Rebuttle**
>
> We sincerely appreciate your careful review and insightful comments.
>
> The VAE paradigm consists of a reconstruction term combined with the KL divergence that regularizes the encoded latent distribution toward a Gaussian prior. Reconstruction is then performed through sampling.
>
> VAE forces the latent space to be "normalized" via KL divergence, which disrupts the distribution and topology of the dimensionality-reduced data because the original data distribution is not necessarily Gaussian, as the datasets SPHERES and RINGS used in our paper.
>
> Our method and TAE do not assume a Gaussian distribution for the dimensionality-reduced data. Furthermore, VAE does not compare the similarity of the distributions before and after dimensionality reduction; it cannot maintain distribution similarity like our method, nor maintain topological similarity like TAE. Therefore, VAEs are not suitable for the comparison of dimensionality reduction.

---

### Meta-Review · Area_Chair_stvp · 2026-01-05

**Summary:**

I am excited about improving dimensionality reduction methods with additional "inductive biases," as proposed by the authors. Defining distributional similarities between _different_ spaces is a great idea that I would love to see more in practice. That being said, I cannot endorse this submission for publication as it stands right now based on the following concerns:

1. Lack of comparisons with _other_ autoencoder / dimensionality reduction methods (reviewer `znS5`).
2. Missing details on what type of topological features, if any, are preserved (reviewer `q3AN`).
3. Missing details on computational complexity (reviewer `q3AN`).
4. Missing contextualization of novelty in light of methods like MMD (reviewer `VJJu`).
5. Missing theoretical analysis (reviewer `bpEu`).

This submission has merits and I encourage the authors to incorporate the feedback. In particular the comments by reviewer `VJJu` on the overall _necessity_ of all proposed components strike me as a great way to better analyze the "practical" performance of the work. While not explicitly mentioned by reviewers, it might also be a good idea to include datasets from life-sciences applications in order to showcase the capabilities of the proposed method better.

**Reviewer Concerns:**

Unfortunately, I find the rebuttal to scarcely address _any_ of the concerns by reviewers, expect for concern 3 (computational complexity). The main issues, viz., lack of comparisons, contextualization, and improved analyses, are _not_ addressed in the rebuttal. Out of these, the missing theoretical analysis could also be addressed by rewriting the paper and toning done some claims; the submission is currently making some strong claims, which are not sufficiently backed up.

**Reviewer Scores:**

Given that the rebuttal did not address the main points of several reviewers (for example, the discussion on why VAEs are _not_ included strikes me as somewhat preliminary; I would have at least expected such a discussion to be in the main text, since VAEs are such a staple in  ML), I find that _none_ of the reviewers would have substantially shifted their scores. This holds in particular for reviewers `znS5` and `VJJu`, who outright stated that the rebuttal was not able to address their concerns.

---

### Decision · Program_Chairs · 2026-01-26

Reject